# Acute Myocardial Infarction among Hospitalizations for Heat Stroke in the United States

**DOI:** 10.3390/jcm9051357

**Published:** 2020-05-06

**Authors:** Tarun Bathini, Charat Thongprayoon, Api Chewcharat, Tananchai Petnak, Wisit Cheungpasitporn, Boonphiphop Boonpheng, Narut Prasitlumkum, Ronpichai Chokesuwattanaskul, Saraschandra Vallabhajosyula, Wisit Kaewput

**Affiliations:** 1Department of Internal Medicine, University of Arizona, Tucson, AZ 85721, USA; 2Division of Nephrology and Hypertension, Department of Medicine, Mayo Clinic, Rochester, MN 55905, USA; api.che@hotmail.com; 3Division of Pulmonary and Critical Care Medicine, Faculty of Medicine Ramathibodi Hospital, Mahidol University, Bangkok 10400, Thailand; petnak@yahoo.com; 4Division of Nephrology, Department of Internal Medicine, University of Mississippi Medical Center, Jackson, MS 39216, USA; 5Department of Medicine, University of California, Los Angeles, Los Angeles, CA 90095, USA; boonpipop.b@gmail.com; 6Department of Medicine, University of Hawaii, Honolulu, HI 96822, USA; narutpra@hawaii.edu; 7Faculty of Medicine, King Chulalongkorn Memorial Hospital, Chulalongkorn University, Bangkok 10330, Thailand; dr_ronpichai_c@yahoo.com; 8Department of Cardiovascular Medicine, Mayo Clinic, Rochester, MN 55905, USA; Vallabhajosyula.Saraschandra@mayo.edu; 9Department of Military and Community Medicine, Phramongkutklao College of Medicine, Bangkok 10400, Thailand

**Keywords:** myocardial infarction, outcomes, hospitalization, heat stroke, resource utilization

## Abstract

Background: This study aimed to assess the risk factors and impact of acute myocardial infarction on in-hospital treatments, complications, outcomes, and resource utilization in hospitalized patients for heat stroke in the United States. Methods: Hospitalized patients with a principal diagnosis of heat stroke were identified in the National Inpatient Sample dataset from the years 2003 to 2014. Acute myocardial infarction was identified using the hospital International Classification of Diseases, Ninth Revision (ICD-9), diagnosis of 410.xx. Clinical characteristics, in-hospital treatment, complications, outcomes, and resource utilization between patients with and without acute myocardial infarction were compared. Results: A total of 3372 heat stroke patients were included in the analysis. Of these, acute myocardial infarction occurred in 225 (7%) admissions. Acute myocardial infarction occurred more commonly in obese female patients with a history of chronic kidney disease, but less often in male patients aged <20 years with a history of hypothyroidism. The need for mechanical ventilation, blood transfusion, and renal replacement therapy were higher in patients with acute myocardial infarction. Acute myocardial infarction was associated with rhabdomyolysis, metabolic acidosis, sepsis, gastrointestinal bleeding, ventricular arrhythmia or cardiac arrest, renal failure, respiratory failure, circulatory failure, liver failure, neurological failure, and hematologic failure. Patients with acute myocardial infarction had 5.2-times greater odds of in-hospital mortality than those without myocardial infarction. The length of hospital stay and hospitalization cost were also higher when an acute myocardial infarction occurred while hospitalized. Conclusion: Acute myocardial infarction was associated with worse outcomes and higher economic burden among patients hospitalized for heat stroke. Obesity and chronic kidney disease were associated with increased risk of acute myocardial infarction, while young male patients and hypothyroidism were associated with decreased risk.

## 1. Introduction 

Heat stroke is a hazardous and potentially life-threatening form of heat-related illness characterized by the presence of an increasing core body temperature higher than 40.5 °C [1,2,3]. It is accompanied by neurological dysfunction such as confusion, seizure, and an alteration of consciousness [4]. Heat stroke is mainly classified into two subcategories, exertional heat stroke and non-exertional heat stroke, also known as “classic” heat stroke [5]. Exertional heat stroke commonly occurs among young healthy persons who are vigorously active in environments with high ambient temperatures and humidity levels [6,7]. By contrast, classic heat stroke is frequently found among older patients with underlying medical conditions that impair their ability to dissipate heat and intervene in their access to hydration [8,9,10].

Under normal conditions, when patients are exposed to a hot environment, the cardiovascular system enhances cardiac output, aiming to increase blood flow to the skin for cooling [11]. This defense mechanism increases the oxygen consumption of the heart and might potentially precipitate a myocardial infarction. Therefore, the association between high temperature and ischemic heart disease has been investigated in recent past years. A previous study demonstrated that higher environmental temperatures are associated with an increase in the risk of myocardial infarction [12]. However, this effect is only influential during the first six hours after exposure [12]. Furthermore, the study previously showed that heat stroke was associated with a higher risk of subsequent acute myocardial infarction and higher mortality [13]. 

Consequently, identifying risk factors of acute myocardial infarction among heat stroke patients who might be a potentially high risk for the condition, aiming to detect it early and begin treatment, might reduce serious complications and mortality. However, knowledge of risk factors and the impact of acute myocardial infarction among hospitalized heat stroke patients are limited and remain unclear. 

Therefore, we conducted this study with the aim of assessing the risk factors and impact of acute myocardial infarction on in-hospital treatments, complications, outcomes, and resource use among patients hospitalized for heat stroke in the United States.

## 2. Materials and Methods

### 2.1. Data Source

This cohort study was conducted using the 2003–2014 National Inpatient Sample (NIS) database. The NIS is the largest all-payer inpatient database in the United States. Discharge dataset from a 20% stratified sample of hospitals in United States with the patient encounter-level information, which includes principal and secondary diagnosis codes as well as procedure codes, are recorded in the NIS. Sample weight is used to generate national estimates for hospitalization nationwide. The approval from the institutional review board was exempted as the information was obtained from a de-identified public database. This study adhered to the policy for protection of human subjects according to Declaration of Helsinki. 

### 2.2. Study Population

Our study included all patients who were admitted in hospitals from 2003 to 2014 with a principal diagnosis of heat stroke, based on International Classification of Diseases, Ninth Revision, Clinical Modification (ICD-9 CM) diagnosis code of 992.0. Heat stroke patients who developed acute myocardial infarction were identified using ICD-9 diagnosis of 410.xx.

### 2.3. Data Collection

Patient characteristics, treatments, complication, and outcomes during hospitalization were identified using ICD-9 codes (Appendix A). Patient characteristics included age, sex, race, smoking, alcohol drinking, obesity, diabetes mellitus, hypertension, hypothyroidism, chronic kidney disease, coronary artery disease, congestive heart failure, and atrial flutter/fibrillation. Treatments included invasive mechanical ventilation, blood component transfusion, and renal replacement therapy. Complications and outcomes included electrolyte derangements (hyponatremia, hypernatremia, hypokalemia, hyperkalemia, hypocalcemia, hypercalcemia, metabolic acidosis, and metabolic alkalosis), rhabdomyolysis, sepsis, gastrointestinal bleeding, ventricular arrhythmia or cardiac arrest, and end-organ failure (renal failure, respiratory failure, liver failure, neurological failure, and hematological failure), and in-hospital mortality. Resource utilization included length of hospital stay and hospitalization cost. 

### 2.4. Statistical Analysis

The total number of heat stroke patients was estimated using discharge-level weights provided by the Healthcare Cost and Utilization Project (HCUP). Continuous variables were reported as mean ± standard deviation and were compared using student’s t-test. Categorical variables were presented as counts with percentage and were compared using Chi-squared test. Multivariable logistic regression was performed to assess if there were clinical characteristics independently associated with acute myocardial infarction. The association of acute myocardial infarction with in-hospital treatments, complications, and outcomes was assessed using logistic regression analysis, and with length of hospital stay and hospitalization cost using linear regression analysis, with pre-specified adjustment for age, sex, smoking, alcohol drinking, and comorbidities. All analyses were two-tailed. Statistical significance was achieved when *p*-value < 0.05. SPSS statistical software (version 22.0, IBM Corporation, Armonk, NY, USA) was used for all analyses. 

## 3. Results

### 3.1. Incidence of and Risk Factors for Acute Myocardial Infarction in Hospitalized Heat Stroke Patients

A total of 3372 hospitalized heat stroke patients were included in analysis. Of these, 225 (7%) had acute myocardial infarction in hospital. Table 1 compared clinical characteristics, in-hospital treatments, complications, outcomes, and resource utilization between heat stroke patients with and without acute myocardial infarction. Patients with acute myocardial infarction were older, were more likely to be female, but less likely to be smokers, and had more comorbidities, including obesity, diabetes mellitus, hypertension, chronic kidney disease, coronary artery disease, and congestive heart failure, compared with patients without acute myocardial infarction.

In multivariable analysis, obesity (OR 1.78; *p* = 0.01) and history of chronic kidney disease (OR 1.61; *p* = 0.04) were independently associated with increased risk of acute myocardial infarction, whereas age < 20 (OR 0.27; *p* = 0.01), male sex (OR 0.58; *p* < 0.001), and history of hypothyroidism (OR 0.33; *p* = 0.005) were associated with decreased risk of acute myocardial infarction (Table 2).

### 3.2. The Association of Acute Myocardial Infarction with In-Hospital Treatments, Complications, and Outcomes

In terms of in-hospital treatments, acute myocardial infarction was significantly associated with higher requirement of invasive mechanical ventilation (OR 4.53; *p* < 0.001), blood component transfusion (OR 4.57; *p* < 0.001), and renal replacement therapy (OR 2.32; *p* = 0.02). Eighteen (8%) and one (0.4%) of the patients with acute myocardial infarction underwent coronary angiogram ± percutaneous coronary intervention and thrombolytic therapy, whereas none of the patients without acute myocardial infarction received these treatments. In term of in-hospital complications and outcomes, acute myocardial infarction was significantly associated with rhabdomyolysis (OR 1.87; *p* < 0.001), metabolic acidosis (OR 2.22; *p* < 0.001), sepsis (OR 3.31; *p* < 0.001), gastrointestinal bleeding (OR 10.95; *p* < 0.001), ventricular arrhythmia/cardiac arrest (OR 3.07; *p* < 0.001), renal failure (OR 1.85; *p* < 0.001), respiratory failure (OR 4.64; *p* < 0.001), circulatory failure (OR 1.79; *p* = 0.002), liver failure (OR 5.48; *p* < 0.001), neurological failure (OR 1.96; *p* < 0.001), and hematological failure (OR 5.21; *p* < 0.001). In addition, acute myocardial infarction was significantly associated with increased in-hospital mortality (OR 5.21; *p* < 0.001) (Table 3).

In-hospital mortality was 0.3%, 1.0%, 5.2%, and 15.6% for patients with 0, 1, 2, and ≥3 in-hospital complications, respectively (*p* < 0.001). There was no interaction between acute myocardial infarction and these in-hospital complications on in-hospital mortality (all *p*-interaction > 0.05).

### 3.3. Impact of Acute Myocardial Infarction on Resource Utilization

Acute myocardial infarction was associated with increased mean length of stay by 3.9 (95% CI 3.0–4.8) days and increased mean hospitalization cost by $41,321 (95% CI 31,465–51,177) (Table 3).

## 4. Discussion

This large cohort study of 3372 patients found that acute myocardial infarction occurred in 7% of patients hospitalized for heat stroke. We identified that obesity and chronic kidney disease were independently associated with increased risk of in-hospital acute myocardial infarction, while age <20 years, male sex, and hypothyroidism were independently associated with decreased risk. Heat stroke patients with in-hospital acute myocardial infarction had higher in-hospital mortality, and greater incidence of several serious complications, and required more resource utilization, compared to patients without acute myocardial infarction.

The impact of acute myocardial infarction includes an increase in in-hospital mortality, incidence of several serious complications, and resource use [14]. Heat stroke patients with acute myocardial infarction were five times more likely to die in hospital than patients without it. In addition to its mortality rate, acute myocardial infarction is strongly associated with gastrointestinal bleeding among heat stroke patients. The rates of mortality and gastrointestinal bleeding among acute myocardial infarction in our study were higher than other cohorts of patients with acute myocardial infarction, suggesting the high impact of acute myocardial infarction on patients hospitalized for heat stroke [15,16,17,18]. Our study also suggested that patients with acute myocardial infarction hospitalized for heat stroke were more likely to develop concomitant multiorgan failure and other complications, including rhabdomyolysis, metabolic acidosis, sepsis, gastrointestinal bleeding, ventricular arrhythmia, or cardiac arrest. In addition, the economic burden is one of the issues that should also be noted. The length of hospital stay and hospitalization costs were higher when an acute myocardial infarction occurred while hospitalized for heat stroke. The hospitalization costs of patients complicated with acute myocardial infarction were more than two times higher than those without it. This could potentially be due to higher concomitant multiorgan failure and other complications requiring more invasive mechanical ventilator and renal replacement therapy among hospitalized patients for heat stroke with acute myocardial infarction (AMI).

The hemodynamic response to a hot environment increases both the function of the heart and oxygen consumption, leading to a risk of myocardial infarction. In our study, we demonstrated that obesity and chronic kidney disease are important risk factors for in-hospital acute myocardial infarction among patients hospitalized for heat stroke, consistent with findings from Wang et al. [13]. This is supported by the fact that chronic kidney disease and obesity are risk factors of cardiovascular diseases [19,20]. While smoking, diabetes mellitus, and hypertension are well known major risk factors of myocardial infarction [21,22], they were not demonstrated at a statistically significant level in multivariable analysis in our study. Although the underlying explanation remains unclear, it is possible that myocardial infarction among patients with heat stroke is type 2 myocardial infarction that occurs secondary to an acute imbalance in myocardial oxygen supply and demand without atherothrombosis [23,24]. Catecholamines induced type 2 myocardial injury/infarction among patients with heat stroke have been reported [23,24,25], and it is possible that the risk for type 2 myocardial infarction among patients with heat stroke might be decreased among males and patients with hypothyroid due to the lesser degree of catecholamine release [26,27]. Previous studies have also shown that patients with type 2 myocardial infarction are older and more often females [21,28], which are consistent with the findings of decreased risk of heat stroke associated myocardial infarction among young male patients in our study.

Several limitations should be pinpointed in our study. First, the NIS is a cohort of hospitalized patients. We expect that the need for hospitalization generally indicates more severe cases of heat stroke or heat stroke in high-risk populations. Therefore, we should be cautious of selection bias. The result might not be generalizable to all heat stroke patients. Second, given the lack of timing of acute myocardial infarction and other complications during hospitalization, we cannot establish their temporal relationship. Therefore, the association between acute myocardial infarction and other in-hospital complications might be bi-directional. Third, the diagnosis of acute myocardial infarction in this study was not based on symptoms, electrocardiogram, cardiac injury biomarkers, or coronary angiogram, but we used the diagnosis code due to the nature of the NIS database. Therefore, the diagnosis of acute myocardial infarction can be heterogeneous and potentially inaccurate. The timing of symptom onset and treatment for revascularization was also lacking. Additionally, the diagnosis code did not provide a sub-classification of heat stroke to indicate whether it was exertional or non-exertional heat stroke, which have different characteristics and severity. Fourth, given the administrative nature of the dataset, the data on medication treatment were limited in this study. Thus, future studies are needed to assess whether the aggressive treatment with the thrombolytic, anticoagulant, and antiplatelet agents accompanied by hematological failure in heat stroke patients might explain this bleeding-related complication. Finally, we could only demonstrate the short-term complications and outcomes during their hospitalizations. Long-term outcomes of acute myocardial infarction after hospital discharge among heat stroke patients were not reported. Nevertheless, our study’s strength is worth mentioning. Our study is the largest observational study to date, including 3372 heatstroke patients of various ages, ethnicities, and underlying diseases.

## 5. Conclusions

In summary, acute myocardial infarction occurred in 7% of patients hospitalized for heat stroke, but its impact should be recognized. Acute myocardial infarction was associated with higher mortality, and occurrence of several serious complications, particularly gastrointestinal bleeding, and organ failures. Obesity and chronic kidney disease were associated with higher risk of acute myocardial infarction, while teenage youth, male sex, and hypothyroidism were associated with lower risk.

## Figures and Tables

**Table 1 jcm-09-01357-t001:** Clinical characteristics, in-hospital treatments, complications, outcomes, and resource utilization in heat stroke patients.

	Total	Acute Myocardial Infarction	No Acute Myocardial Infarction	*p*-Value
Clinical characteristics				
N (%)	3372	225 (6.7)	3147 (93.3)	
Age (years)	55 ± 22	62 ± 20	54 ± 22	<0.001
<20	218 (6.5)	4 (1.8)	214 (6.8)	<0.001
20–39	654 (19.4)	27 (12.0)	627 (19.9)	
40–59	1034 (30.7)	69 (30.7)	965 (30.7)	
60–79	900 (26.7)	65 (28.9)	835 (26.6)	
≥80	564 (16.7)	60 (26.7)	504 (16.0)	
Sex				<0.001
Male	2478 (73.5)	135 (60.0)	2343 (74.5)	
Female	894 (26.5%)	90 (40.0)	804 (25.5%)	
Race				0.67
Caucasian	1883 (55.8)	118 (52.4)	1765 (56.1)	
African American	496 (14.7)	33 (14.7)	463 (14.7)	
Hispanic	428 (12.7)	33 (14.7)	395 (12.6)	
Other	565 (16.8)	41 (18.2)	524 (16.7)	
Smoking	604 (17.9)	22 (9.8)	582 (18.5)	0.001
Alcohol drinking	270 (8.0)	13 (5.8)	257 (8.2)	0.20
Obesity	233 (6.9)	26 (11.6)	207 (6.6)	0.004
Diabetes Mellitus	562 (16.7)	56 (24.9)	506 (16.1)	0.001
Hypertension	1255 (37.2)	112 (49.8)	1143 (36.3)	<0.001
Dyslipidemia	495 (14.7)	41 (18.2)	454 (14.4)	0.12
Hypothyroidism	196 (5.8)	7 (3.1)	189 (6.0)	0.07
Chronic kidney disease	201 (6.0)	24 (10.7)	177 (5.6)	0.002
Coronary artery disease	389 (11.5)	37 (16.4)	352 (11.2)	0.02
Congestive heart failure	216 (6.4)	23 (10.2)	193 (6.1)	0.02
Atrial flutter/fibrillation	251 (7.4)	22 (9.8)	229 (7.3)	0.17
Treatment				
Invasive mechanical ventilation	686 (20.3)	94 (41.8)	592 (18.8)	<0.001
Blood component transfusion	169 (5.0)	35 (15.6)	134 (4.3)	<0.001
Renal replacement therapy	75 (2.2)	11 (4.9)	64 (2.0)	0.005
Coronary angiogram ± primary coronary intervention	18 (0.5)	18 (8.0%)	0 (0)	N/A
Complication and outcomes				
Hyponatremia	293 (8.7)	14 (6.2)	279 (8.9)	0.17
Hypernatremia	183 (5.4)	16 (7.1)	167 (5.3)	0.25
Hypokalemia	500 (14.8)	32 (14.2)	468 (14.9)	0.79
Hyperkalemia	134 (4.0)	10 (4.4)	124 (3.9)	0.71
Hypocalcemia	73 (2.2)	3 (1.3)	70 (2.2)	0.38
Hypercalcemia	37 (1.1)	1 (0.4)	36 (1.1)	0.33
Rhabdomyolysis	1049 (31.1)	83 (36.9)	966 (30.7)	0.05
Metabolic acidosis	472 (14.0)	50 (22.2)	422 (13.4)	<0.001
Metabolic alkalosis	29 (0.9)	4 (1.8)	25 (0.8)	0.12
Sepsis	149 (4.4)	29 (12.9)	120 (3.8)	<0.001
Gastrointestinal bleeding	54 (1.6)	16 (7.1)	38 (1.2)	<0.001
Ventricular arrhythmia/cardiac arrest	95 (2.8)	16 (7.1)	79 (2.5)	0.001
Renal failure	1226 (36.4)	103 (45.8)	1123 (35.7)	0.002
Respiratory failure	550 (16.3)	87 (38.7)	463 (14.7)	<0.001
Circulatory failure	393 (11.7)	40 (17.8)	353 (11.2)	0.003
Liver failure	196 (5.8)	35 (15.6)	161 (5.1)	<0.001
Neurological failure	651 (19.3)	68 (30.2)	583 (18.5)	<0.001
Hematological failure	449 (13.3)	60 (26.7)	389 (12.4)	<0.001
In-hospital mortality	168 (5.0)	35 (15.6)	133 (4.2)	<0.001
Resource utilization				
Length of hospital stay (days)	4.3 ± 7.0	8.0 ± 13.4	4.0 ± 6.2	<0.001
Hospitalization cost ($)	35,335 ± 72,085	72,444 ± 117,246	32,757 ± 67,112	<0.001

Continuous variables were reported as mean ± standard deviation; categorical variables as counts (percentages).

**Table 2 jcm-09-01357-t002:** Univariable and multivariable analysis assessing factors associated with acute myocardial infarction in heat stroke patients.

Variables	Univariable Analysis		Multivariable Analysis	
	Crude Odds Ratio (95% CI)	*p*-Value	Adjusted Odds Ratio (95% CI)	*p*-Value
Age (years)				
<20	0.26 (0.09–0.72)	0.01	0.27 (0.10–0.76)	0.01
20–39	0.60 (0.38–0.95)	0.03	0.70 (0.43–1.12)	0.13
40–59	1 (reference)		1 (reference)	
60–79	1.09 (0.77–1.55)	0.64	0.91 (0.62–1.33)	0.62
≥80	1.67 (1.16–2.39)	0.006	1.35 (0.89–2.04)	0.16
Male	0.51 (0.39–0.68)	<0.001	0.58 (0.43–0.79)	<0.001
Race				
Caucasian	1 (reference)		1 (reference)	
African American	1.07 (0.72–1.59)	0.75	0.98 (0.65–1.48)	0.91
Hispanic	1.25 (0.84–1.87)	0.28	1.40 (0.92–2.12)	0.12
Other	1.17 (0.81–1.69)	0.40	1.20 (0.82–1.73)	0.37
Smoking	0.48 (0.31–0.75)	0.001	0.68 (0.41–1.10)	0.10
Alcohol drinking	0.69 (0.39–1.23)	0.21	0.89 (0.49–1.62)	0.70
Obesity	1.86 (1.20–2.86)	0.005	1.78 (1.13–2.80)	0.01
Diabetes Mellitus	1.73 (1.26–2.37)	0.001	1.21 (0.85–1.72)	0.28
Hypertension	1.74 (1.33–2.28)	<0.001	1.28 (0.93–1.75)	0.13
Dyslipidemia	1.32 (0.93–1.88)	0.12	1.07 (0.73–1.58)	0.73
Hypothyroidism	0.50 (0.23–1.08)	0.08	0.33 (0.15–0.71)	0.005
Chronic kidney disease	2.00 (1.28–3.14)	0.002	1.61 (1.01–2.56)	0.04
Coronary artery disease	1.56 (1.08–2.26)	0.02	1.22 (0.81–1.82)	0.35
Congestive heart failure	1.74 (1.11–2.75)	0.02	1.18 (0.71–1.95)	0.52
Atrial flutter/fibrillation	1.38 (0.87–2.19)	0.17	0.95 (0.57–1.57)	0.83

**Table 3 jcm-09-01357-t003:** The association between acute myocardial infarction and in-hospital treatments, complications, outcomes, and resource utilization in heat stroke patients.

	Univariable Analysis		Multivariable Analysis	
	Crude Odds Ratio (95% CI)	*p*-Value	Adjusted Odds Ratio * (95% CI)	*p*-Value
Treatments				
Invasive mechanical ventilation	3.10 (2.34–4.10)	<0.001	4.53 (3.29–6.24)	<0.001
Blood component transfusion	4.14 (2.78–6.18)	<0.001	4.57 (2.97–7.04)	<0.001
Renal replacement therapy	2.48 (1.29–4.77)	0.007	2.32 (1.13–4.77)	0.02
Complications and outcomes				
Hyponatremia	0.68 (0.39–1.19)	0.18	0.68 (0.39–1.20)	0.18
Hypernatremia	1.37 (0.80–2.33)	0.25	1.37 (0.80–2.36)	0.26
Hypokalemia	0.95 (0.65–1.40)	0.79	0.96 (0.64–1.42)	0.82
Hyperkalemia	1.13 (0.59–2.19)	0.71	1.11 (0.57–2.17)	0.76
Hypocalcemia	0.59 (0.19–1.90)	0.38	0.64 (0.20–2.09)	0.46
Hypercalcemia	0.39 (0.05–2.83)	0.39	0.58 (0.08–4.38)	0.60
Rhabdomyolysis	1.32 (0.99–1.75)	0.05	1.87 (1.37–2.53)	<0.001
Metabolic acidosis	1.85 (1.33–2.57)	<0.001	2.22 (1.57–3.15)	<0.001
Metabolic alkalosis	2.26 (0.78–6.55)	0.13	1.97 (0.65–5.92)	0.23
Sepsis	3.73 (2.43–5.74)	<0.001	3.31 (2.11–5.21)	<0.001
Gastrointestinal bleeding	6.26 (3.44–11.42)	<0.001	10.95 (3.37–35.50)	<0.001
Ventricular arrhythmia/cardiac arrest	2.97 (1.71–5.18)	<0.001	3.07 (1.71–5.48)	<0.001
Renal failure	1.52 (1.16–2.00)	0.003	1.85 (1.38–2.47)	<0.001
Respiratory failure	3.66 (2.75–4.86)	<0.001	4.64 (3.39–6.35)	<0.001
Circulatory failure	1.71 (1.20–2.45)	0.003	1.79 (1.24–2.59)	0.002
Liver failure	3.42 (2.30–5.07)	<0.001	5.48 (3.50–8.59)	<0.001
Neurological failure	1.91 (1.41–2.57)	<0.001	1.96 (1.45–2.67)	<0.001
Hematological failure	2.58 (1.88–3.53)	<0.001	3.22 (2.28–4.55)	<0.001
In-hospital mortality	4.17 (2.79–6.22)	<0.001	5.21 (3.37–8.06)	<0.001
	Coefficient (95% CI)	*p*-value	Adjusted coefficient (95% CI)	*p*-value
Resource utilization				
Length of hospital stay (days)	3.9 (3.8–4.3)	<0.001	3.9 (3.0–4.8)	<0.001
Hospitalization cost ($)	39,687 (29,832–49,541)	<0.001	41,321 (31,465–51,177)	<0.001

* Adjusted for age, sex, race, smoking, alcohol drinking, obesity, diabetes mellitus, hypertension, dyslipidemia, hypothyroidism, chronic kidney disease, chronic ischemic heart disease, congestive heart failure, and atrial flutter/fibrillation.

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
