# Peer review of "Acute Myocardial Infarction among Hospitalizations for Heat Stroke in the United States"

_jcm, 2020, doi:10.3390/jcm9051357_

Round 1
Reviewer 1 Report
Bathini et al. performed the study to examine the incidence and impact of acute myocardial infarction on in-hospital clinical outcomes, complications, treatments, and resource use in patients hospitalized for heat stroke in the United States. A total of 3,372 patients who were hospitalized for heat stroke from 2003 to 2014 were included in the analysis. The data for the analysis were acquired from the 2003-2014 National Inpatient Sample database. The authors found that acute myocardial infarction occurred in 7% of patients hospitalized for heat stroke. They demonstrated that the patients with acute myocardial infarction were older and had greater prevalence of comorbidities such as obesity, diabetes, hypertension, chronic kidney disease, coronary artery disease, and congestive heart failure. They identified that obesity and chronic kidney disease were independently associated with increased risk, while age <20 years, male sex, and hypothyroidism with decreased risk of in-hospital acute myocardial infarction in patients hospitalized for heat stroke. The investigators also found that the heat stroke patients with in-hospital acute myocardial infarction had higher mortality and greater incidence of serious complications during hospitalization compared to the patients without myocardial infarction.
This study addresses an interesting topic and the rationale and scientific content of the manuscript are valuable. However, I have several concerns that, in my opinion, deserve special attention in the revision.
- The Results section needs improvements, especially more details are needed. Some parts of the tables lack the data and it is not clear why. Here are just a few examples; the lack of the number of females with and without myocardial infarction in Table 1, and the lack of p values and odds ratios in Table 1 and Table 2. It is unclear why the odds ratios and p values for smoking or diabetes are lacking in multivariable analysis (Table 2).
- In my view more detailed description of results in the Results section is needed, including more specific comments about the most important findings. For example, there are no comments about the data on the comparison between the groups of patients with and without acute myocardial infarction which are shown in Table 1. However, some of these results are interesting and are worthy of emphasis, e.g. the information that the patients with acute myocardial infarction were less frequently smokers than those without, etc.
- In the Results section, the comments regarding the results of multivariable analysis appear to be inaccurate. Specifically, based on Table 2 the obesity (p=0.005) and chronic kidney disease (p=0.04) are independently associated with increased risk of myocardial infarction; however hypertension is not (p=0.06). In addition, the age <20 years (p=0.01) and male sex (<0.001) should be included as the factors that are associated with decreased risk. There is also some problem with the current interpretation of old age for which the p values indicate statistical insignificance rather than significance.
- The Discussion section also needs improvements. Some additions and corrections related to my comments above are needed. In addition, the important complications such as gastrointestinal bleeding or sepsis are missing in the sentence: Rhabdomyolysis, metabolic acidosis, and multiorgan failures were also associated with acute myocardial infarction. Moreover, this statement has been constructed ambiguously, so the clarification is needed. The language and style also require special attention and improvements. Here is one simple example how relatively minor edits result in significant improvement of a sentence. The current version of the sentence (line 160) is Heat stroke patients complicated with the coronary syndrome were five times more likely than patients without it to experience a myocardial infarction. I suggest: Heat stroke patients with complications of coronary syndrome were five times more likely to experience a myocardial infarction than patients without it. In addition, the terms such as acute myocardial infarction, coronary disease, and acute coronary syndrome, are used in the text interchangeably, which is incorrect. In addition, the data about pharmacotherapy are not included in the Results section, so it is inappropriate to include them in the Discussion. These data (or references to these data) should be added in the Results. Finally, I suggest improving or enhancing the discussion of original findings of this study in the broader context of relevant results from other studies. I also suggest making an attempt to indicate potential mechanisms which might impact the most significant findings of this study. I think that the description of the potential mechanisms which is provided in the fourth paragraph of the Discussion is insufficient. Also, there are other relevant references that can be included, e.g. Wallace et al., Prior heat illness hospitalization and risk of early death, Environmental Research, 2007.
- The Abstract and Conclusions section also require further attention and some improvements according to my comments about Results and Discussion sections.
- The scientific writing requires further attention because some statements can be made clearer to provide better understanding of specific message or content, especially in the Results and Discussion sections. The language and style also require checking and minor improvements.
- Please address the question if the investigators adhered to policies for protection of human subjects because IRB approval was exempted.
Author Response
Response to Reviewer#1
Comment
Bathini et al. performed the study to examine the incidence and impact of acute myocardial infarction on in-hospital clinical outcomes, complications, treatments, and resource use in patients hospitalized for heat stroke in the United States. A total of 3,372 patients who were hospitalized for heat stroke from 2003 to 2014 were included in the analysis. The data for the analysis were acquired from the 2003-2014 National Inpatient Sample database. The authors found that acute myocardial infarction occurred in 7% of patients hospitalized for heat stroke. They demonstrated that the patients with acute myocardial infarction were older and had greater prevalence of comorbidities such as obesity, diabetes, hypertension, chronic kidney disease, coronary artery disease, and congestive heart failure. They identified that obesity and chronic kidney disease were independently associated with increased risk, while age <20 years, male sex, and hypothyroidism with decreased risk of in-hospital acute myocardial infarction in patients hospitalized for heat stroke. The investigators also found that the heat stroke patients with in-hospital acute myocardial infarction had higher mortality and greater incidence of serious complications during hospitalization compared to the patients without myocardial infarction.
This study addresses an interesting topic and the rationale and scientific content of the manuscript are valuable. However, I have several concerns that, in my opinion, deserve special attention in the revision
Response: We thank you for reviewing our manuscript and for your critical evaluation. We really appreciated your input and found your suggestions very helpful.
Comment #1
The Results section needs improvements, especially more details are needed. Some parts of the tables lack the data and it is not clear why. Here are just a few examples; the lack of the number of females with and without myocardial infarction in Table 1, and the lack of p values and odds ratios in Table 1 and Table 2. It is unclear why the odds ratios and p values for smoking or diabetes are lacking in multivariable analysis (Table 2).
Response: The reviewer raised very important point. The number of females with and without acute myocardial infarction and p-value for each category of race and age groups were added in revised Table 1 as suggested.
Previously, we reported the final multivariable model which was selected using forward stepwise selection. Therefore, odds ratio of those variables which did not retain in the model was not reported. Odds ratio and p-value when including all variables in multivariable model were reported in Table 2 as suggested.
Comment #2
In my view more detailed description of results in the Results section is needed, including more specific comments about the most important findings. For example, there are no comments about the data on the comparison between the groups of patients with and without acute myocardial infarction which are shown in Table 1. However, some of these results are interesting and are worthy of emphasis, e.g. the information that the patients with acute myocardial infarction were less frequently smokers than those without, etc.
Response: We appreciate the reviewer’s comments. We agree and the following statements have been added to the result section.
Patients with acute myocardial infarction were older, were more likely to be female, but less likely to be smokers, and had more comorbidities, including obesity, diabetes mellitus, hypertension, chronic kidney disease, coronary artery disease, and congestive heart failure, compared with patients without acute myocardial infarction.
Comment #3
In the Results section, the comments regarding the results of multivariable analysis appear to be inaccurate. Specifically, based on Table 2 the obesity (p=0.005) and chronic kidney disease (p=0.04) are independently associated with increased risk of myocardial infarction; however hypertension is not (p=0.06). In addition, the age <20 years (p=0.01) and male sex (<0.001) should be included as the factors that are associated with decreased risk. There is also some problem with the current interpretation of old age for which the p values indicate statistical insignificance rather than significance.
Response: We apologized for this errors. The following statements have been revised to be more accurate.
In multivariable analysis, obesity (OR 1.78; p=0.01), and history of chronic kidney disease (OR 1.61; p=0.04) were independently associated with increased risk of acute myocardial infarction, whereas age <20 (OR 0.27; p=0.01), male sex (OR 0.58; p<0.001), and history of hypothyroidism (OR 0.33; p=0.005) were associated with decreased risk of acute myocardial infarction (Table 2).
Comment #4
The Discussion section also needs improvements. Some additions and corrections related to my comments above are needed. In addition, the important complications such as gastrointestinal bleeding or sepsis are missing in the sentence: Rhabdomyolysis, metabolic acidosis, and multiorgan failures were also associated with acute myocardial infarction. Moreover, this statement has been constructed ambiguously, so the clarification is needed. The language and style also require special attention and improvements. Here is one simple example how relatively minor edits result in significant improvement of a sentence. The current version of the sentence (line 160) is Heat stroke patients complicated with the coronary syndrome were five times more likely than patients without it to experience a myocardial infarction. I suggest: Heat stroke patients with complications of coronary syndrome were five times more likely to experience a myocardial infarction than patients without it. In addition, the terms such as acute myocardial infarction, coronary disease, and acute coronary syndrome, are used in the text interchangeably, which is incorrect. In addition, the data about pharmacotherapy are not included in the Results section, so it is inappropriate to include them in the Discussion. These data (or references to these data) should be added in the Results. Finally, I suggest improving or enhancing the discussion of original findings of this study in the broader context of relevant results from other studies. I also suggest making an attempt to indicate potential mechanisms which might impact the most significant findings of this study. I think that the description of the potential mechanisms which is provided in the fourth paragraph of the Discussion is insufficient.
Also, there are other relevant references that can be included, e.g. Wallace et al., Prior heat illness hospitalization and risk of early death, Environmental Research, 2007.
Response: We appreciate the reviewer’s thorough and helpful review and we agree with the reviewer. We have reviewed our manuscript carefully again and revised our discussion as reviewer’s suggestions. We agree with the reviewer that the terms acute myocardial infarction and acute coronary syndrome are interchangeable. We have made corrections to follow the correct terms according to ICD diagnosis as suggested.
We have also removed the discussion on pharmacotherapy as suggested. We added this as limitation of our study. “Fourth, given the administrative nature of the dataset, the data on medication treatment were limited in this study. Thus, future studies are needed to assess whether the aggressive treatment with the thrombolytic, anticoagulant, and antiplatelet agents accompanied by hematological failure in heat stroke patients might explain this bleeding-related complication.”
We have also revised our discussion and provided more discussions on potential mechanisms as the reviewer’s suggestion.
The suggested literature is very helpful and relevant to our study. We have also added Wallace et al., Prior heat illness hospitalization and risk of early death, Environmental Research, 2007 as new reference (3)
Comment #5
The Abstract and Conclusions section also require further attention and some improvements according to my comments about Results and Discussion sections
Response: The reviewer raises very important points. We incoperated all important suggestions from the reviewer and revised our manuscript throughtout as suggested.
Comment #6
The scientific writing requires further attention because some statements can be made clearer to provide better understanding of specific message or content, especially in the Results and Discussion sections. The language and style also require checking and minor improvements
Response: We agree with the reviewer. We have comprehensively reviewed our manuscript again and revised throughout manuscript to improve our manuscript as reviewer’s suggestions.
Comment #7
Please address the question if the investigators adhered to policies for protection of human subjects because IRB approval was exempted.
Response: We agree with the reviewer. The following statements have been added to the method section
“This study adhered to the policy for protection of human subjects according to Declaration of Helsinki.”
We greatly appreciated the editor and reviewer’s time and comments to improve our manuscript.

Reviewer 2 Report
In this study, the authors aimed of assessing the risk factors and impact of acute myocardial infarction (AMI) on in-hospital treatments, complications, outcomes, and resource use among 3,372 patients hospitalized for heat stroke in the United States. The study focuses on an interesting population, data are well presented and findings are intriguing. I offer the following comments:
1) In this specific setting, the authors should specify that type 2 AMI is probably the most frequent AMI type occurring during heat stroke. Indeed, in this report, AMI was more frequently associated with rhabdomyolysis, metabolic acidosis, sepsis, gastrointestinal bleeding, renal failure, and respiratory failure. This supports that AMI, in this clinical context, is not usually due to atherothrombosis. Rather, all these complications, which can be ascribed to the ongoing heat stroke, may be the cause of AMI. Thus, no causal-effect link between heat stroke and AMI can be inferred. This point should be properly clarified by the authors. Moreover, to further support this hypothesis, the presence of known coronary artery disease, diabetes mellitus, and smoking habit were not associated with an increased risk of AMI.
2) In the text, the authors use indiscriminately the terms acute coronary syndrome and acute myocardial infarction. Please, refer to acute myocardial infarction only.
3) Data on invasive mechanical ventilation, blood transfusion, renal replacement therapy, gastrointestinal bleeding, and sepsis have been collected. Can the authors show the number of patients with only one of these complications, two, and so on? And it would of interest to show how mortality rises as the number of complications increases. Finally, the authors should investigate how AMI interacts with these data (a p for interaction may be of interest)
4) Multivariable analyses have been performed and the authors provide adjusted OR. It is not clear which variables have been chosen for adjustment. Please, clarify this issue and clarify whether variables were adjusted for all each others.
5) AMI is associated with an increased mean hospitalization cost. This can be due to the more frequent need of mechanical ventilation, RRT, and so on that are not necessarily associated with AMI. Please, address this point.
Author Response
Response to Reviewer#2
Comment
In this study, the authors aimed of assessing the risk factors and impact of acute myocardial infarction (AMI) on in-hospital treatments, complications, outcomes, and resource use among 3,372 patients hospitalized for heat stroke in the United States. The study focuses on an interesting population, data are well presented and findings are intriguing. I offer the following comments:
Response: We thank you for reviewing our manuscript and for your critical evaluation. We really appreciated your input and found your suggestions very helpful.
Comment #1
In this specific setting, the authors should specify that type 2 AMI is probably the most frequent AMI type occurring during heat stroke. Indeed, in this report, AMI was more frequently associated with rhabdomyolysis, metabolic acidosis, sepsis, gastrointestinal bleeding, renal failure, and respiratory failure. This supports that AMI, in this clinical context, is not usually due to atherothrombosis. Rather, all these complications, which can be ascribed to the ongoing heat stroke, may be the cause of AMI. Thus, no causal-effect link between heat stroke and AMI can be inferred. This point should be properly clarified by the authors. Moreover, to further support this hypothesis, the presence of known coronary artery disease, diabetes mellitus, and smoking habit were not associated with an increased risk of AMI.
Response: We greatly appreciate the reviewer’s comment. The reviewer’s comment is very helpful and help us explain the possible explainations underlying the findings of our study. We agree and have included the discussion of type 2 AMI in our discussion as reviewer’s suggestion. The following texts have been added to the discussion.
“While smoking, diabetes mellitus, and hypertension are well known major risk factors of myocardial infarction (19, 20), they were not demonstrated at a statistically significant level in multivariable analysis in our study. Although the underlying explanation remains unclear, it is possible that myocardial infarction among patients with heat stroke is type 2 myocardial infarction that occurs secondary to an acute imbalance in myocardial oxygen supply and demand without atherothrombosis (21, 22). Catecholamines induce type 2 myocardial injury/infarction among patients with heat stroke have been reported (21-23), and it is possible that the risk for type 2 myocardial infarction among patients with heat stroke might be decreased among males and patients with hypothyroid due to the lesser degree of catecholamine release (24, 25). Previous studies have also shown that patients with type 2 myocardial infarction are older and more often females (19, 26), which are consistent with the findings of decreased risk of heat stroke associated myocardial infarction among young male patients in our study.”
Comment #2
In the text, the authors use indiscriminately the terms acute coronary syndrome and acute myocardial infarction. Please, refer to acute myocardial infarction only.
Response: We appreciate the reviewer’s thorough and helpful review and we agree with the reviewer. We have reviewed our manuscript carefully again and revised our discussion as reviewer’s suggestions. We agree with the reviewer that the terms acute myocardial infarction and acute coronary syndrome are interchangeable. We have made corrections to follow the correct terms according to ICD diagnosis as suggested.
Comment #3
Data on invasive mechanical ventilation, blood transfusion, renal replacement therapy, gastrointestinal bleeding, and sepsis have been collected. Can the authors show the number of patients with only one of these complications, two, and so on? And it would of interest to show how mortality rises as the number of complications increases. Finally, the authors should investigate how AMI interacts with these data (a p for interaction may be of interest)
Response: The following table showed in-hospital mortality rate based on the number of in-hospital complications.
|
Number of complications |
Total |
In-hospital mortality |
|
0 |
1051 |
3 (0.3) |
|
1 |
933 |
9 (1) |
|
2 |
579 |
30 (5) |
|
>= 3 |
809 |
126 (16) |
The following table showed the p-interaction between acute myocardial infarction and other in-hospital complications on in-hospital mortality
|
Variable |
P-interaction |
|
Hyponatremia |
0.02 |
|
Hypernatremia |
0.009 |
|
Hypokalemia |
0.71 |
|
Hyperkalemia |
0.36 |
|
Hypocalcemia |
0.25 |
|
Hypercalcemia |
0.70 |
|
Rhabdomyolysis |
0.50 |
|
Metabolic acidosis |
0.22 |
|
Metabolic alkalosis |
0.32 |
|
Sepsis |
0.23 |
|
Gastrointestinal bleeding |
0.77 |
|
Ventricular arrhythmia/cardiac arrest |
0.09 |
|
Renal failure |
0.35 |
|
Respiratory failure |
0.08 |
|
Circulatory failure |
0.25 |
|
Liver failure |
0.45 |
|
Neurological failure |
0.14 |
|
Hematological failure |
0.26 |
The following statements have been added to the result section as suggested.
In-hospital mortality was 0.3%, 1.0%, 5.2%, 15.6% for patients with 0, 1, 2, and ≥3 in-hospital complications (p<0.001). There was no interaction between acute myocardial infarction and these in-hospital complications on in-hospital mortality (all p-interaction >0.05).
Comment #4
Multivariable analyses have been performed and the authors provide adjusted OR. It is not clear which variables have been chosen for adjustment. Please, clarify this issue and clarify whether variables were adjusted for all each others.
Response: We agree with the reviewer. We revised analysis, as suggested by reviewer 1, to include all baseline characteristics in the multivariable model, as shown in Table 2. Therefore, the odds ratio were adjusted for all each others.
Comment #5
AMI is associated with an increased mean hospitalization cost. This can be due to the more frequent need of mechanical ventilation, RRT, and so on that are not necessarily associated with AMI. Please, address this point.
Response: We appreciate the reviewer’s comment. We agree and we have added this discussion as the reviewer’s suggestion. The following text has been added.
The hospitalization costs of patients complicated with acute myocardial infarction were more than two times higher than those without it. This could potentially be due to higher concomitant multiorgan failure and other complications requiring more invasive mechanical ventilator and renal replacement therapy among hospitalized patients for heat stroke with AMI.
We greatly appreciated the editor and reviewer’s time and comments to improve our manuscript.

Round 2
Reviewer 2 Report
All my criticisms have been satisfied. I have no further comments.
Author Response
We thank you for reviewing our manuscript and for your critical evaluation. We really appreciated your input and found your suggestions very helpful.
In addition, the following statements have been added to the result section.
“18 (8%) and 1 (0.4%) of patients with acute myocardial infarction underwent coronary angiogram ± percutaneous coronary intervention and thrombolytic therapy, whereas none of patients without no acute myocardial infarction received these treatments.”
However, as the diagnosis of acute myocardial infarction was based on diagnosis code, the database did not contain the timing of symptom onset and the treatment for revascularization. The following statements have been added to the limitation section.
“The timing of symptom onset and treatment for revascularization was also lacking.”